# Deep Unsupervised Hashing via External Guidance

**Qihong Song** [1 2]  **Xiting Liu** [3]  **Hongyuan Zhu** [4]  **Joey Tianyi Zhou** [5 6]  **Xi Peng** [1 7]  **Peng Hu** [1 2]

## Abstract

Recently, deep unsupervised hashing has gained considerable attention in image retrieval due to its advantages in cost-free data labeling, computational efficiency, and storage savings. Although existing methods achieve promising performance by leveraging inherent visual structures within the data, they primarily focus on learning discriminative features from unlabeled images through limited internal knowledge, resulting in an intrinsic upper bound on their performance. To break through this intrinsic limitation, we propose a novel method, called Deep Unsupervised Hashing with External Guidance (DUH-EG), which incorporates external textual knowledge as semantic guidance to enhance discrete representation learning. Specifically, our DUH-EG: i) selects representative semantic nouns from an external textual database by minimizing their redundancy, then matches images with them to extract more discriminative external features; and ii) presents a novel bidirectional contrastive learning mechanism to maximize agreement between hash codes in internal and external spaces, thereby capturing discrimination from both external and intrinsic structures in Hamming space. Extensive experiments on four benchmark datasets demonstrate that our DUH-EG remarkably outperforms existing state-of-the-art hashing methods. The code is available at: https://github.com/XLearning-SCU/2025-ICML-DUHEG

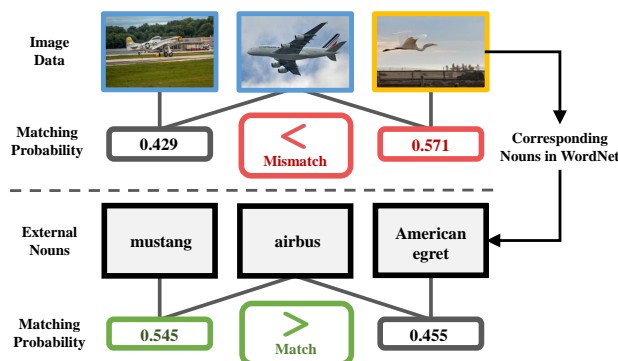

*Figure 1.* The matching result among three images from the MSCOCO dataset. The category of the two images above (with a blue border) is the airplane, while the remaining one (with a yellow border) is categorized as a bird. We illustrate the matching probability based on features of image data and external nouns extracted by the CLIP (ViT-B/16) model (Radford et al., 2021), respectively. Specifically, for two images with a higher matching probability, if they belong to different categories (indicating a mismatch), their matching probability is marked in red. Otherwise, the probability is marked in green.

## 1. Introduction

High retrieval efficiency and low storage cost are essential for large-scale information retrieval tasks. Hashing methods map high-dimensional data into compact binary codes, significantly reducing storage space and accelerating computation through XOR operation, while preserving the discrimination in the discrete representations. As a result, it has received increasing attention from both academia and industry. Existing hashing methods can be broadly categorized into supervised and unsupervised hashing. While supervised hashing methods achieve promising performance with well-labeled data, large-scale data annotation is labor-intensive, costly, and even infeasible in practice. By contrast, unsupervised approaches leverage unlabeled data to learn binary representations, thereby avoiding cost-prohibitive data annotation and drawing more practical interest. The key to unsupervised hashing lies in constructing intrinsic data structures without class labels.

To address this challenge, most unsupervised hashing methods explore mining inherent information from unlabeled data to facilitate hash code learning, such as data reconstruction (Do et al., 2016; Song et al., 2018; Zieba et al., 2018),

---

[1]The College of Computer Science, Sichuan University Chengdu 610065, China [2]The State Key Laboratory of Integrated Services Network, Xidian University, Xi'an 710071, China [3]Georgia Institute of Technology, USA [4]I[2]R & CFAR, A*STAR [5]Centre for Frontier AI Research (CFAR), Agency for Science, Technology and Research (A*STAR), Singapore [6] Institute of High Performance Computing, Agency for Science, Technology and Research (A*STAR), Singapore [7]National Key Laboratory of Fundamental Algorithms and Models for Engineering Numerical Simulation, Sichuan University, China. Correspondence to: Peng Hu <penghu.ml@gmail.com>.

*Proceedings of the 42nd International Conference on Machine Learning*, Vancouver, Canada. PMLR 267, 2025. Copyright 2025 by the author(s).

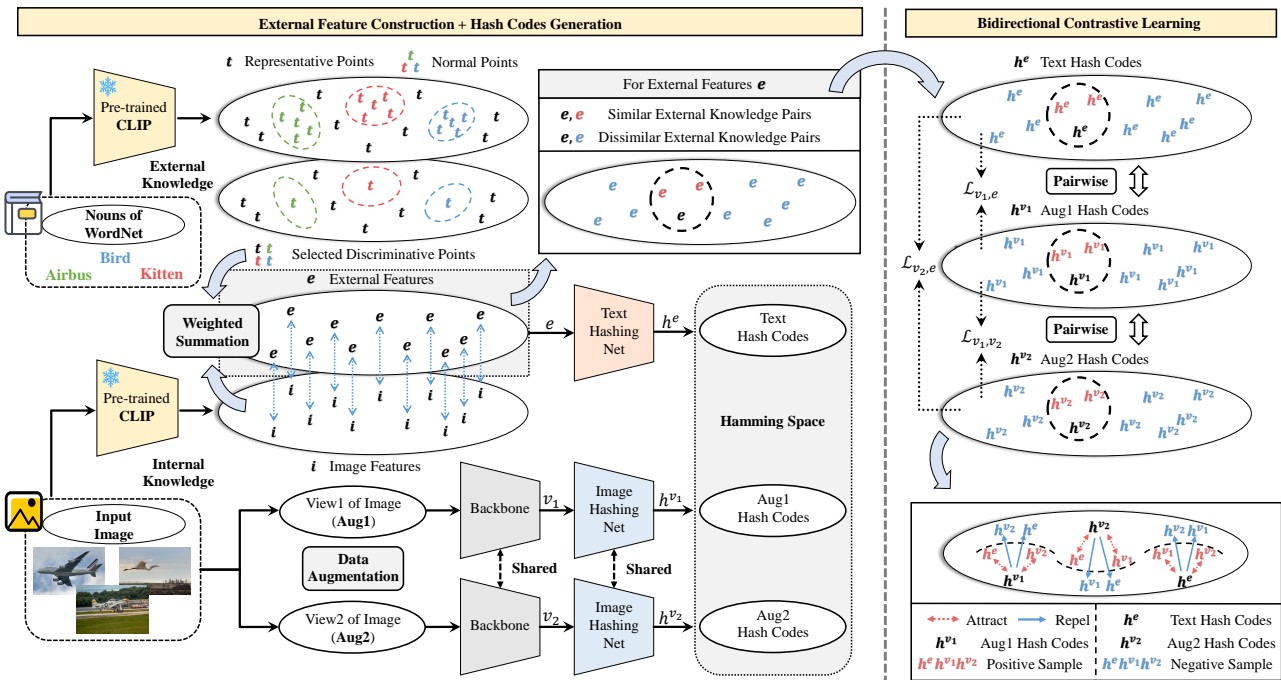

*Figure 2.* The overall framework of the proposed DUH-EG method. To be specific, in the left part, we construct the external features $e$ corresponding to the visual feature $i$ of input images and identify similar (positive) and dissimilar (negative) external knowledge pairs, marking them with different colors. Then, we generate hash codes of the external feature $h^e$ and two augmented views $h^{v_1}$ and $h^{v_2}$, respectively. After that, we employ a bidirectional contrastive learning loss to maximize the agreement across the positive samples as shown in the right part. Note that we use distinct colors to represent different clusters of normal points in the left part.

similarity preserving (Tu et al., 2020; Shen et al., 2018; Yang et al., 2019; Huang et al., 2016; Shen et al., 2019; Gu et al., 2019), contrastive learning (Luo et al., 2021b; Lin et al., 2022; Yu et al., 2022), etc. Data reconstruction methods typically utilize generative models to learn binary bottleneck representations by reconstructing input data, encouraging hash codes to preserve as much intrinsic information from input data as possible (Dai et al., 2017; Shen et al., 2020; Cao et al., 2018). However, such methods may preserve some non-discriminative information for reconstruction in the Hamming space, such as extraneous background, leading to a sacrifice in hash code discriminability (Qiu et al., 2021). In contrast, similarity-preserving methods (Luo et al., 2021a; Dong et al., 2020) employ pre-trained models or clustering techniques to extract underlining similarity structures, such as pairwise similarity (Yang et al., 2018; Ma et al., 2024) or pseudo-labels (Hu et al., 2017; Zhang et al., 2017), from unlabeled inputs to guide hash function learning, thus enhancing the discriminability of the discrete representations. Inspired by the great success achieved by contrastive learning (Chen et al., 2020), recent studies (Qiu et al., 2021; Ma et al., 2022; Luo et al., 2021b; Lin et al., 2022; Yu et al., 2022) have integrated it to maximize mutual information between positive pairs in Hamming space, constructed from data augmentation (Qiu et al., 2021), similarity structure (Wei et al., 2024), or pseudo-labels (Luo et al., 2021c), resulting in stronger supervision and more discriminative

hash codes. Despite these advancements, these methods, which rely solely on internal visual structure, may face an inherent limitation on semantic guidance, potentially resulting in suboptimal performance.

To be specific, the exclusive use of internal supervision may introduce semantic ambiguity, thereby leading to an inherent performance ceiling. For instance, as illustrated in Figure 1, we extracted visual features from three images using a pre-trained CLIP (ViT-B/16) model (Radford et al., 2021), and calculated the matching probabilities based on normalized similarity scores. Surprisingly, an airplane image has a higher matching probability with a bird image than with another airplane image (0.571 v.s. 0.429). Fortunately, by associating the images with external nouns (e.g. "airbus", "mustang", and "American_egret"), we achieved clearer distinctions and more accurate matching probabilities (0.545 v.s. 0.455). In brief, leveraging external textual information could thus provide richer and more precise guidance for hash learning, enhancing the discriminability of discrete codes. However, it is challenging to find representative external nouns that discriminatively depict each image without class labels. Intuitively, we could retrieve related nouns from a word database, such as WordNet (Miller, 1995), but excessive synonyms may be introduced and dominate textual semantics, leading to semantic homogeneity and discrimination loss. Additionally, integrating textual infor-

mation as external guidance presents challenges in aligning image-text semantics across different modalities, as well as reconciling internal and external knowledge for consistent semantic guidance.

To address the challenges, we propose Deep Unsupervised Hashing with External Guidance (DUH-EG) to enhance the semantic similarity guidance for hash learning, as shown in Figure 2. Specifically, we first extract text features of each external noun using a pre-trained vision-language model, clustering them to select representative features, thereby mitigating semantic homogeneity caused by excessive synonyms. DUH-EG then extracts image features using the vision-language model, matching them with the representative textual features to integrate external features for each image. To balance internal and external information, we present a bidirectional contrastive learning loss to align textual and visual views for each image while maximizing the agreement across different augmented views. However, traditional contrastive learning roughly treats different images as negative pairs, inevitably sampling some intra-class views as negatives, i.e., false negatives. To eliminate the adverse impact of false negative pairs, we select latent positive pairs from the negatives by using external features, enabling the model to focus on semantic positives rather than only different views of the same sample. The main contributions of this work are summarized as follows:

- We propose a novel approach that incorporates external textual guidance to facilitate discrete representation learning, overcoming the inherent limitations of internal visual structures.

- An efficient, automated, and non-generative method is presented to select representative semantic nouns from an accessible external textual database, integrating the external textual features as external guidance of the corresponding images.

- A bidirectional contrastive learning loss is developed to preserve discrimination in hash codes by maximizing mutual information between semantic positive pairs instead of different views, thereby enhancing effectiveness and robustness.

- Extensive experiments on the four widely-used benchmarks, i.e., CIFAR-10, NUS-WIDE, Flickr25k, and MSCOCO, demonstrate that our approach remarkably outperforms state-of-the-art unsupervised hashing methods.

## 2. Related Work

### 2.1. Deep Unsupervised Hashing

Deep unsupervised hashing approaches can primarily be categorized into data reconstruction (Do et al., 2016; Song et al.,

2018; Zieba et al., 2018) and similarity preserving (Yang et al., 2018; Huang et al., 2016; Shen et al., 2019). For data reconstruction, various studies (Do et al., 2016; Song et al., 2018; Zieba et al., 2018; He et al., 2024) employ generative models such as variational auto-encoders (VAEs) (Kingma & Welling, 2014) and generative adversarial networks (GANs) (Goodfellow et al., 2014) to reconstruct images and produce hash codes that preserve intrinsic information implicit in inputs. For example, (Shen et al., 2020) proposes a Wasserstein auto-encoder variant that uses code-driven adjacency graphs to guide image reconstruction. Similarly, (Cao et al., 2018) leverages images synthesized by Pair Conditional Wasserstein GAN (PC-WGAN) to augment training data, enabling the hash model to learn hash codes that respect semantic pairwise relationships. For similarity preserving, some methods mine semantic similarity structures from pre-trained models to learn similarity-preserving hash codes (Yang et al., 2018; Tu et al., 2020; Shen et al., 2018; Ma et al., 2024). By contrast, other methods leverage clustering techniques to generate pseudo-labels that facilitate similarity-preserving learning in the Hamming space (Huang et al., 2016; Shen et al., 2019; Wei et al., 2023). For instance, (Luo et al., 2021a) employs deep clustering and similarity exploration to build pairwise semantic similarity structures from both local and global perspectives. Likewise, (Qiu et al., 2024) applies hierarchical clustering in the Euclidean tangent space to extract pseudo hierarchical semantics, thereby providing additional training supervision. Recently, contrastive learning has been introduced to learn hash representations by maximizing instance discrimination, achieving promising performance (Luo et al., 2021b; Yu et al., 2022; Wei et al., 2024). However, almost all of these methods seek to explore the intrinsic information implicit in the input data, leading to an inherent performance ceiling.

*Improvement*: Differing from existing similarity-preserving methods, our DUH-EG introduces the corresponding textual features as external knowledge for each image used in training with an efficient, automated, and non-generative approach. Furthermore, rather than generating pseudo-labels, DUH-EG balances the internal and external knowledge of the images by leveraging a contrastive learning strategy that aligns the visual and textual features of each training image. Additionally, textual features are utilized to identify potential false-negative pairs, thereby maximizing the consistency of features within the same class.

### 2.2. Learning with External Guidance

In recent years, there has been a growing interest in incorporating external guidance to enhance model performance. External resources, such as noun phrases (Li et al., 2024), co-occurrence relations of entities (Yang et al., 2023), instance-aware information (Yan et al., 2023), and large language models (Wang et al., 2023), have been increasingly used

to enrich understanding and improve reasoning capabilities. This typically involves reusing existing guidance and integrating external guidance to uncover latent and implicit representations, providing more comprehensive guidance information and improving model performance. For example, in the task of learning with noisy labels, (Wang et al., 2023) utilize large language models to assess the categories of training samples and generate confidence scores, which guide the fine-tuning of pre-trained language models. In 3D scene generation, (Wu et al., 2024) extract the relative positional relationships and associated probabilities between the paired objects from existing indoor scene data. They incorporate this external guidance to mitigate the ambiguity in object shapes and layouts in sketches, ultimately enhancing the diversity of the generated content. Similarly, in image classification and object detection, (Shen et al., 2022) select noun phrases from external textual sources, such as category names and text titles, and append them to the original text data. These augmented textual inputs enhance the visual conceptual information in the data, improving the ability of the model to recognize and classify objects. Although these methods have demonstrated promising results with external guidance, their effectiveness in the context of hash learning remains an open question.

## 3. Method

### 3.1. Overview

Let $\mathcal{X} = \{x_i\}_{i=1}^{N_i}$ represents an unlabeled dataset that contains $\{N_i\}$ image samples. The goal of deep unsupervised hashing is to learn a hash function $\mathcal{H} : x_i \to h_i \in \{-1, +1\}^L$ that encodes the $i$-th sample of images $x_i$ as $L$-bit compact discrete representation $h_i$ for efficient image retrieval. Following a common practice in most previous hashing methods (Qiu et al., 2021; Ma et al., 2024; Luo et al., 2021b), we assume that the hash codes are composed of $\pm 1$, i.e., $h_i \in \{-1, +1\}^L$.

The core challenge of unsupervised hashing is to construct semantic guidance from input data without relying on semantic labels. However, the semantic guidance extracted solely from the images is constrained by their inherent visual structure. To overcome this limitation, we propose DUH-EG, which introduces external knowledge to improve the semantic similarity guidance for hash learning. To illustrate our DUH-EG, we categorize this approach into two distinct phases: External Feature Construction (Section 3.2) and Bidirectional Contrastive Learning (Section 3.3). In the first phase, the proposed method screens representative external nouns and extracts external features that match well with the visual features. For the subsequent stage, we present a bidirectional contrastive learning loss to maximize agreement between textual and visual views of the same image, thus encapsulating the instance discrimination into

hash codes. Simultaneously, we eliminate false negatives by selecting latent positive pairs using external features, enabling the model to focus on semantic positives. In addition, for ease of differentiation, we denote the aligned external feature and two differently augmented views of $x_i$ as $e_i$, $v_i^1$ and $v_i^2$, respectively. The hashing networks applied to the augmented views and external features as $\mathcal{H}_v$ and $\mathcal{H}_e$ respectively, where $\mathcal{H}_v$ consists of a pre-trained backbone followed by a hashing layer, while $\mathcal{H}_e$ consists solely of a hashing layer to project the external features into $L$-bit hash codes. In our DUH-EG, the outputs of each hash function are defined as $h_i^{v_1} = \mathcal{H}_v(v_i^1, \Theta_v)$, $h_i^{v_2} = \mathcal{H}_v(v_i^2, \Theta_v)$ and $h_i^e = \mathcal{H}_e(e_i, \Theta_e)$ where $\Theta_v$ and $\Theta_e$ are the learnable parameters in the hashing networks. $h_i^{v_1}$, $h_i^{v_2}$ and $h_i^e$ are $L$-bit hash codes ($\{-1, +1\}^L$) corresponding to two augmented views and an external textual view, respectively. Thus, to maximize agreement across each view of the redefined positive pairs, the bidirectional contrastive learning loss function is formulated as:

$$\mathcal{L}_{BCL} = \mathcal{L}_b(h_i^{v_1}, h_i^{v_2}) + \mathcal{L}_b(h_i^{v_1}, h_i^e) + \mathcal{L}_b(h_i^{v_2}, h_i^e). \quad (1)$$

### 3.2. External Feature Construction

In this phase, we first select all nouns from WordNet (Miller, 1995) as an external noun set. Following a prompt template subset recommended by CLIP [1] (contains 7 templates in total), each noun is then constructed into several natural phrases $\{text_w^p\}_{p=1}^{N_p}$, where $w$ corresponds to the $w$-th sample in the noun set, $p$ refers to the index of prompt templates, and $N_p$ is the number of prompt templates. Subsequently, we utilize a pre-trained CLIP model (Radford et al., 2021) to extract textual features $t_w^p$ for each constructed phrase. To integrate the textual features of each word into a unified representation, we fuse these features through

$$\bar{t}_w = \frac{1}{N_p} \sum_{p=1}^{N_p} t_w^p \quad w \in \{1, \cdots, N_t\}, \quad (2)$$

where $N_t$ indicates the number of samples in the noun set. Next, to select representative semantic nouns that discriminatively characterize each image, we apply the K-means clustering to the fused textual features and generate $N_k$ clusters. For the $k$-th cluster, we select textual features that are relatively distant from the cluster center $c_k$ and treat them as a representative subset $\{\bar{t}_w^{rep}\}_{w=1}^{N_{rep}}$ with $N_{rep}$ features, while the remaining features are denoted as $\{\bar{t}_w^{nor}\}_{w=1}^{N_{nor}}$.

$$\begin{cases} \{\bar{t}_{w,k}\} \in \{\bar{t}_w^{rep}\} & \text{if } dis(\bar{t}_{w,k}, c_k) > T_1, \\ \{\bar{t}_{w,k}\} \in \{\bar{t}_w^{nor}\} & \text{if } dis(\bar{t}_{w,k}, c_k) \leq T_1, \end{cases} \quad (3)$$
$$\{\bar{t}_w\}_{w=1}^{N_t} = \{\{\bar{t}_w^{rep}\}_{w=1}^{N_{rep}}, \{\bar{t}_w^{nor}\}_{w=1}^{N_{nor}}\},$$

---

[1] https://github.com/openai/CLIP/blob/main/notebooks/Prompt_Engineering_for_ImageNet.ipynb

where $\bar{t}_{w,k}$ indicates that $\bar{t}_w$ belongs to the $k$-th cluster, $dis(\bar{t}_{w,k}, c_k)$ denotes the distance between each word and its cluster center in the public embedding space, and $T_1$ is a distance threshold. In addition, to preserve informative features of each cluster center, we predict the softmax probability $p_{w,k}$ that each remaining word $\bar{t}_w^{nor}$ belongs to a cluster center $c_k$ as

$$p_{w,k} = \frac{exp(cos(\bar{t}_w^{nor}, c_k))}{\sum_{j=1}^{N_k} exp(cos(\bar{t}_w^{nor}, c_j))}. \tag{4}$$

For each cluster, we select textual features with the highest softmax probability relative to the cluster center to represent it, forming a new subset $\{\bar{t}_w^{cen}\}_{w=1}^{N_{cen}}$. We combine $\{\bar{t}_w^{rep}\}_{w=1}^{N_{rep}}$ and $\{\bar{t}_w^{cen}\}_{w=1}^{N_{cen}}$ to form a final representative subset $\{\bar{t}_w^{com}\}_{w=1}^{N_{com}} = \{\{\bar{t}_w^{rep}\}_{w=1}^{N_{rep}}, \{\bar{t}_w^{cen}\}_{w=1}^{N_{cen}}\}$.

Subsequently, we use the pre-trained CLIP model to extract image features $i_i$ and compute their counterpart from $\{\bar{t}_w^{com}\}_{w=1}^{N_{com}}$ for each image data $x_i$. To be specific, we calculate the softmax probability between the image features and each text feature, and treat the probability as a weight. Hence, through applying weighted summation, the external feature $e_i$ corresponding to each image can be calculated as

$$e_i = \sum_{w=1}^{N_{com}} s_{i,w} \bar{t}_w^{com}, \tag{5}$$

$$s_{i,w} = \frac{exp(cos(i_i, \bar{t}_w^{com})/\tau_1)}{\sum_{k=1}^{N_{com}} exp(cos(i_i, \bar{t}_k^{com})/\tau_1)}, \tag{6}$$

where $s_{i,w}$ indicates the softmax probability between the image features $i_i$ and each natural phrase feature $\bar{t}_w^{com}$, and $\tau_1$ is the temperature parameter.

### 3.3. Bidirectional Contrastive Learning

To leverage external features, we propose a bidirectional contrastive learning loss that maximizes mutual information between images and their corresponding textual features. Specifically, we apply random transformations to each image $x_i$, generating two augmented views $v_i^1$ and $v_i^2$, which are aligned with the external feature $e_i$. Next, we project both the augmented views and the external text features into a Hamming space using two distinct hashing networks, $\mathcal{H}_v$ and $\mathcal{H}_e$, producing $L$-bit hash codes: $h_i^{v_1}$, $h_i^{v_2}$ and $h_i^e$. To facilitate the computation of the Hamming distance $dis_H(h_1, h_2)$ between two hash codes, we adopt the inner product: $dis_H(h_1, h_2) = \frac{1}{2}(L - \langle h_1, h_2 \rangle)$. This formulation ensures that the similarity measure, $sim(h_1, h_2) = \frac{1}{2L}(L + \langle h_1, h_2 \rangle)$ is positively correlated with the inner product, allowing it to quantify similarity effectively in the Hamming space.

After generating the hash codes, we employ a bidirectional contrastive learning loss (as illustrated in Figure 2) to align

the discrete representation of textual and visual views for each image while maximizing the mutual information between the two augmented views. However, traditional contrastive learning only treats different views of the same image as positive pairs, as in Equation (7), inevitably neglecting the potential semantic similarity among other samples and misidentifying some intra-class samples as negatives.

$$\mathcal{L}(a,b) = -\frac{1}{N_i} \sum_{i=1}^{N_i} \log \frac{\sum_{j=i} exp(\frac{\langle h_i^a, h_j^b \rangle}{\tau_2})}{\sum_{k=1}^{N_i} exp(\frac{\langle h_i^a, h_k^b \rangle}{\tau_2})}, \tag{7}$$

where $a, b \in \{v_1, v_2, e\}$ ($a \neq b$), and $\tau_2$ is a temperature parameter controlling the scaling of similarity.

To reduce the detrimental effect of false negative pairs, we redefine positive and negative pairs by finding external feature pairs with relatively high similarity. To be specific, we first treat the hash codes of external features ($h_i^e$) and two augmented views ($h_i^{v_1}$ and $h_i^{v_2}$) that are derived from the same image as positive sample pairs. After that, to select potential positive pairs from the negatives, we calculate the cosine similarity between all external features and determine the semantic positive and negative pairs as

$$\begin{cases} j \in \{i_{neg}\} & \text{if } cos(e_i, e_j) \leq T_2, \\ j \in \{i_{pos}\} & \text{if } cos(e_i, e_j) > T_2, \\ \{i_{neg}\} \cup \{i_{pos}\} = \{1, 2, \cdots, N_i\}, \end{cases} \tag{8}$$

where $T_2$ indicates a similarity threshold. If the similarity between a sample pair $e_i$ and $e_j$ exceeds this threshold, we consider the $j$-th image $x_j$ as a positive sample pair ($j \in \{i_{pos}\}$) corresponding to the $i$-th image $x_i$. Conversely, if the similarity does not exceed the threshold, the image pair is considered as a negative sample pair ($j \in \{i_{neg}\}$). Based on the redefined positive pairs, we modify Equation (7) to

$$\mathcal{L}_b(a,b) = -\frac{1}{N_i} \sum_{i=1}^{N_i} \log \frac{\sum_{j \in \{i_{pos}\}} exp(\frac{\langle h_i^a, h_j^b \rangle}{\tau_2})}{\sum_{k=1}^{N_i} exp(\frac{\langle h_i^a, h_k^b \rangle}{\tau_2})}. \tag{9}$$

## 4. Experiments

### 4.1. Experiment Setup

#### 4.1.1. DATASETS

- **CIFAR-10** (Krizhevsky & Hinton, 2009) - Dataset contains 60,000 photographs divided into ten categories, with 6,000 images in each class. Following the protocol of previous studies (Qiu et al., 2021; Wang et al., 2022; Qiu et al., 2024), we randomly pick 1,000 photographs from each class as the query set, for a total of 10,000 images. We take the remaining images as the retrieval set and randomly choose 500 images per class as the training set from the retrieval set.

| Method | References | CIFAR-10 16 bits 32 bits 64 bits | | | NUS-WIDE 16 bits 32 bits 64 bits | | | Flickr25k 16 bits 32 bits 64 bits | | | MSCOCO 16 bits 32 bits 64 bits | | |
|---|---|---|---|---|---|---|---|---|---|---|---|---|---|
| **CNN** | | | | | | | | | | | | | |
| CIBHash (Qiu et al., 2021) | IJCAI21 | 0.590 | 0.622 | 0.641 | 0.790 | 0.807 | 0.815 | - | - | - | 0.737 | 0.760 | 0.775 |
| DATE (Luo et al., 2021b) | ACMMM21 | 0.577 | 0.629 | 0.647 | 0.798 | 0.810 | 0.815 | 0.822 | 0.841 | 0.844 | - | - | - |
| MeCoQ (Wang et al., 2022) | AAAI22 | 0.629 | 0.641 | 0.651 | 0.802 | 0.822 | 0.832 | 0.813 | 0.817 | 0.827 | - | - | - |
| DSCH (Lin et al., 2022) | AAAI22 | 0.624 | 0.644 | 0.670 | 0.762 | 0.780 | 0.786 | 0.817 | 0.827 | 0.828 | - | - | - |
| NSH (Yu et al., 2022) | IJCAI22 | 0.706 | 0.733 | 0.756 | 0.758 | 0.811 | 0.824 | - | - | - | 0.746 | 0.774 | 0.783 |
| CGHash (Song et al., 2023) | ACMMM23 | 0.795 | 0.803 | 0.817 | 0.814 | 0.833 | 0.841 | - | - | - | 0.768 | 0.782 | 0.791 |
| DDCH (Wei et al., 2023) | PR23 | 0.611 | 0.648 | 0.658 | 0.781 | 0.798 | 0.808 | - | - | - | 0.721 | 0.759 | 0.779 |
| HAMAN (Ma et al., 2022) | TCSVT23 | - | - | - | 0.806 | 0.825 | 0.834 | 0.796 | 0.813 | 0.826 | 0.722 | 0.775 | 0.787 |
| HiHPQ (Qiu et al., 2024) | AAAI24 | 0.633 | 0.658 | 0.671 | 0.799 | 0.821 | 0.826 | 0.807 | 0.826 | 0.830 | - | - | - |
| HARR (Ma et al., 2024) | TOMM24 | 0.520 | 0.536 | 0.575 | 0.807 | 0.826 | 0.841 | 0.818 | 0.830 | 0.838 | 0.748 | 0.789 | 0.816 |
| LGH (Zhao et al., 2024) | ICMR24 | 0.846 | 0.862 | 0.874 | 0.815 | 0.828 | 0.837 | - | - | - | 0.796 | 0.814 | 0.827 |
| HHCH(Wei et al., 2024) | TIP24 | 0.631 | 0.657 | 0.681 | 0.797 | 0.820 | 0.828 | 0.825 | 0.838 | 0.842 | 0.775 | 0.798 | 0.810 |
| VGG-16 | **Ours** | 0.887 | 0.888 | 0.885 | 0.852 | 0.854 | 0.856 | 0.856 | 0.858 | 0.856 | 0.833 | 0.840 | 0.842 |
| VGG-16$_{best-T_2}$ | **Ours** | **0.889** | **0.889** | **0.891** | **0.852** | **0.855** | **0.856** | **0.856** | **0.858** | **0.856** | **0.842** | **0.846** | **0.847** |
| **ViT** | | | | | | | | | | | | | |
| UDBH (Guo et al., 2023) | TCSVT23 | 0.775 | 0.779 | 0.783 | 0.829 | 0.840 | 0.849 | 0.845 | 0.850 | 0.860 | - | - | - |
| DDCH (Wei et al., 2023) | PR23 | 0.882 | 0.923 | 0.935 | - | - | - | - | - | - | 0.816 | 0.866 | 0.875 |
| FSCH (Cao et al., 2023) | TCSVT23 | 0.876 | 0.912 | 0.926 | 0.812 | 0.832 | 0.844 | 0.815 | 0.838 | 0.849 | 0.760 | 0.787 | 0.799 |
| *HHCH (Wei et al., 2024) | TIP24 | 0.881 | 0.907 | 0.912 | 0.812 | 0.829 | 0.838 | 0.815 | 0.854 | 0.866 | 0.805 | 0.849 | 0.853 |
| CLIP | **Ours** | 0.931 | 0.934 | 0.933 | 0.849 | 0.855 | 0.855 | 0.874 | 0.887 | 0.892 | 0.854 | 0.878 | 0.887 |
| CLIP$_{best-T_2}$ | **Ours** | **0.939** | **0.940** | **0.940** | **0.849** | **0.856** | **0.856** | **0.874** | **0.887** | **0.892** | **0.862** | **0.881** | **0.888** |

*Table 1.* Mean Average Precision (MAP) comparison among different state-of-the-art deep unsupervised hashing methods on four benchmark datasets with hash code lengths varying from 16 to 64. Note: i) '*' indicates that the source code for the paper is available, and we used the CLIP (ViT-B/16) model as its backbone. ii) The "best-$T_2$" represents the optimal MAP results corresponding to different values of $T_2$ (as illustrated in Figure 5b and indicated with star markers).

- **NUS-WIDE** (Chua et al., 2009) - Dataset includes 269,648 photos in 81 different categories. We concentrate on the top 21 categories following (Zhao et al., 2024; Cao et al., 2023; Ma et al., 2024), from which we randomly select 100 images per category to create a query set (2,100 photographs in total), and the remaining images are served as the retrieval set. Additionally, 10,500 randomly chosen images (500 images per class) from the retrieval set are used as the training set.

- **Flickr25k** (Huiskes & Lew, 2008) - 25,000 multi-label photos with 24 classes are included in the dataset. After eliminating unlabeled data, following the protocol used in previous studies (Ma et al., 2022; Cao et al., 2023; Ma et al., 2024), 2,000 images are randomly chosen as the query set, while the remaining images are utilized as the retrieval set. In the retrieval set, 5,000 images are randomly chosen for training.

- **MSCOCO** (Lin et al., 2014) - 123,287 samples are contained in the dataset, and a subset of 122,218 images from 80 categories is used in our approach. In this subset, similar to (Cao et al., 2023), we randomly select 5,000 images as the query set, and use the remaining images for retrieval, with 10,000 of them further used for training.

#### 4.1.2. IMPLEMENTATION DETAILS

In Section 3.3, we employ a VGG-16 network (Simonyan & Zisserman, 2015) pre-trained on ImageNet-1K and a pre-trained CLIP (ViT-B/16) model (Radford et al., 2021) as the backbone of $\mathcal{H}_v$, respectively. The hashing layer of $\mathcal{H}_v$ consists of a two-layer multi-layer perceptron (MLP) with dimensions $[F-512-L]$, where $F$ represents the output dimension of the pre-trained backbones, while $L$ denotes the length of the generated hash codes, selected from $\{16, 32, 64\}$. To be specific, $F$ was set to 4096 for the pre-trained VGG-16 model, and to 512 for the pre-trained CLIP model. In addition, we adopted a non-cyclic cosine annealing strategy for the learning rate at each epoch $t$.

$$lr_t = lr_{\min} + (lr_{\max} - lr_{\min}) \times lr_{\text{temp}}, \qquad (10)$$

where $lr_{\min}$ indicates the minimum learning rate and $lr_{\max}$ refers to the maximum learning rate. The $lr_{\text{temp}}$ is denoted as:

$$lr_{\text{temp}} = \begin{cases} t/T_{\text{warm}} & t < T_{\text{warm}}, \\ \frac{1}{2}\left(1 + \cos\left(\frac{(t-T_{\text{warm}})\pi}{T_{\max}-T_{\text{warm}}}\right)\right) & t \geq T_{\text{warm}}. \end{cases} \quad (11)$$

Let $T_{\max}$ denotes the total number of epochs and $T_{\text{warm}}$ represents the warm-up epoch. In our training procedure, we fixed $lr_{\min} = 1 \times 10^{-5}$, $lr_{\max} = 1 \times 10^{-4}$, $T_{\max} = 60$, $T_{\text{warm}} = 10$, $N_k = 800$, $T_1 = 0.9$ and $T_2 = 0.97$.

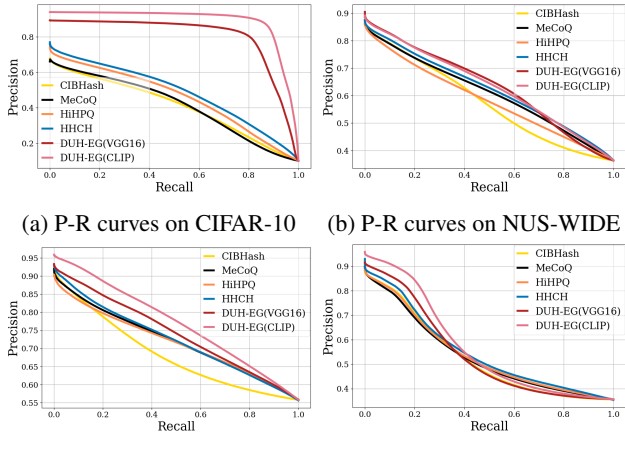

(a) P-R curves on CIFAR-10     (b) P-R curves on NUS-WIDE

(c) P-R curves on Flickr25k     (d) P-R curves on MSCOCO

*Figure 3.* Results of Precision-Recall curves with 64-bit hash codes on four benchmark datasets.

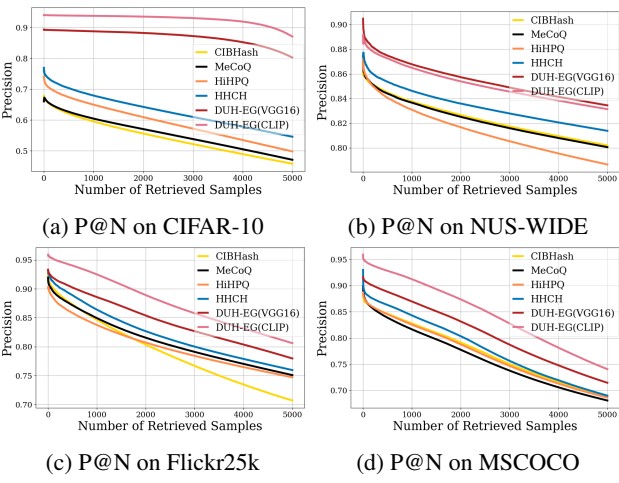

(a) P@N on CIFAR-10     (b) P@N on NUS-WIDE

(c) P@N on Flickr25k     (d) P@N on MSCOCO

*Figure 4.* Results of TopN-Precision curves with 64-bit hash codes on four benchmark datasets.

## 4.2. Comparisons with State-of-The-Art

### 4.2.1. MEAN AVERAGE PRECISION

In this section, we conduct a comparison between DUH-EG and 14 state-of-the-art deep unsupervised hashing approaches on the four image retrieval datasets with hash code lengths varying from 16 to 64. The approaches include CIBHash (Qiu et al., 2021), DATE (Luo et al., 2021b), MeCoQ (Wang et al., 2022), DSCH (Lin et al., 2022), NSH (Yu et al., 2022), CGHash (Song et al., 2023), DDCH (Wei et al., 2023), HAMAN (Ma et al., 2022), HiHPQ (Qiu et al., 2024), HARR (Ma et al., 2024), LGH (Zhao et al., 2024), HHCH (Wei et al., 2024), UDBH (Guo et al., 2023) and FSCH (Cao et al., 2023). To assess the quality of the hash codes that are produced, we employed the standard statistic Mean Average Precision (MAP). In accordance with (Qiu et al., 2021; Ma et al.,

2022; Qiu et al., 2024), we adopt MAP@1000 for CIFAR-10 dataset and MAP@5000 for NUS-WIDE, Flickr25k and MSCOCO datasets while evaluation. Table 1 illustrates the MAP results of our proposed method DUH-EG on four image retrieval datasets with hash code lengths varying from 16 to 64. It is clear that our method consistently achieves the best performance compared with the state-of-the-art methods on four benchmark datasets. To be specific, in the Convolutional Neural Network (CNN) part, we adopt a VGG-16 model pre-trained on ImageNet as the backbone and compare its performance with other baseline models. Our DUH-EG achieves average improvements of 3.39%, 3.03%, 2.53% and 4.04% on CIFAR-10, NUS-WIDE, Flickr25k and MSCOCO datasets compared to the most competitive methods LGH (Zhao et al., 2024), CGHash (Song et al., 2023) and DATE (Luo et al., 2021b), separately. In Vision Transformer (ViT) part, we adopt a pre-trained CLIP (ViT-B/16) model as the backbone and evaluate its performance against other ViT-based methods. Our method achieves an average increase of 2.95%, 1.71%, 4.70% and 2.95% on CIFAR-10, NUS-WIDE, Flickr25k and MSCOCO datasets compared with the top-performing methods DDCH (Wei et al., 2023), UDBH (Guo et al., 2023) and HHCH (Wei et al., 2024), respectively. Moreover, the improvements are particularly evident in the case of a short hash code on the four datasets.

### 4.2.2. PRECISION-RECALL AND TOPN-PRECISION

To evaluate retrieval quality, we follow (Wei et al., 2024; Ma et al., 2024; Cao et al., 2023; Duan et al., 2025) to apply the following additional metrics: i) Precision-Recall (P-R) curves and ii) TopN-Precision (P@N) curves different number of retrieved samples. We report the P-R curves for the 64-bit hash codes in Figure 3. Obviously, DUH-EG outperforms all the compared methods by large margins on CIFAR-10 dataset. Additionally, on NUS-WIDE, Flickr25k and MSCOCO datasets, DUH-EH achieves higher precision at lower recall rates. For P@N curves illustrated in Figure 4, DUH-EG demonstrates significantly superior performance compared to other methods across four benchmark datasets. Notably, on the CIFAR-10 dataset, as the number of retrieval samples increases, DUH-EG exhibits a much slower rate of precision degradation relative to the other methods. This highlights that by incorporating external guidance, our DUH-EG can generate high-quality hash codes.

### 4.3. Ablation Studies

In this section, we conduct ablation studies to validate the indispensability of methods used in DUH-EG on all four benchmark datasets. Specifically, as shown in Table 2, we evaluate the effect of the *SIM* and *AUG* modules while using external features introduced in Section 3.2. Here, the *SIM* module represents utilizing the mechanism illustrated

| | Module | | CIFAR-10 | | | NUS-WIDE | | | Flickr25k | | | MSCOCO | | |
| | SIM | AUG | 16 bits | 32 bits | 64 bits | 16 bits | 32 bits | 64 bits | 16 bits | 32 bits | 64 bits | 16 bits | 32 bits | 64 bits |
|---|---|---|---|---|---|---|---|---|---|---|---|---|---|---|
| #1 | - | - | 0.922 | 0.926 | 0.925 | 0.827 | 0.833 | 0.835 | 0.845 | 0.860 | 0.871 | 0.812 | 0.849 | 0.864 |
| #2 | ✓ | - | 0.922 | 0.925 | 0.925 | 0.832 | 0.838 | 0.839 | 0.859 | 0.870 | 0.878 | 0.855 | 0.874 | 0.883 |
| #3 | - | ✓ | 0.939 | 0.940 | 0.940 | 0.848 | 0.854 | 0.855 | 0.864 | 0.880 | 0.886 | 0.831 | 0.862 | 0.874 |
| #4 | ✓ | ✓ | **0.939** | **0.940** | **0.940** | **0.849** | **0.856** | **0.856** | **0.874** | **0.887** | **0.892** | **0.862** | **0.881** | **0.888** |

*Table 2.* Ablation studies on the *SIM* and *AUG* modules with different bit lengths across four benchmark datasets.

| External | CIFAR-10 | | | NUS-WIDE | | | Flickr25k | | | MSCOCO | | |
| Knowledge | 16 bits | 32 bits | 64 bits | 16 bits | 32 bits | 64 bits | 16 bits | 32 bits | 64 bits | 16 bits | 32 bits | 64 bits |
|---|---|---|---|---|---|---|---|---|---|---|---|---|
| *IMGAUG* | 0.922 | 0.923 | 0.929 | 0.828 | 0.839 | 0.844 | 0.839 | 0.856 | 0.866 | 0.733 | 0.785 | 0.819 |
| *IMGK* | 0.924 | 0.927 | 0.929 | 0.836 | 0.846 | 0.846 | 0.840 | 0.857 | 0.864 | 0.739 | 0.794 | 0.820 |
| *EK* | 0.935 | 0.936 | 0.937 | 0.845 | 0.850 | 0.851 | 0.844 | 0.852 | 0.858 | 0.858 | 0.877 | 0.883 |
| *IMGCLS* | 0.934 | 0.937 | 0.936 | 0.838 | 0.848 | 0.849 | 0.842 | 0.859 | 0.867 | 0.748 | 0.798 | 0.823 |
| *EFC* | **0.939** | **0.940** | **0.940** | **0.849** | **0.856** | **0.856** | **0.874** | **0.887** | **0.892** | **0.862** | **0.881** | **0.888** |

*Table 3.* Ablation studies on the *IMGAUG*, *IMGK*, *IMGCLS*, *EK* and *EFC* methods with different bit lengths across four benchmark datasets.

in Equation (8) to identify the potential positive pairs and negative pairs in bidirectional contrastive learning loss by the similarity threshold ($T_2$). Moreover, the *AUG* module refers to incorporating two augmented views of images in bidirectional contrastive learning loss and maximizing mutual information between the discrete representation of these two views. Table 2 illustrates the MAP result of different configurations on four benchmark datasets. In particular, when considering module *SIM* and *AUG* individually, the comparison results of (*#1* v.s. *#2*) demonstrate a notable contribution of the *SIM* module to performance improvement. Especially on the MSCOCO dataset, the performance of the 16-bit hash codes increases with a gain of $5.30\%$. For the *AUG* module, comparing the MAP result between *#1* and *#3* clearly indicates that enabling the *AUG* module leads to a significant enhancement in performance across all four datasets. In addition, configuration *#4* demonstrates that the simultaneous introduction of both the *SIM* and *AUG* modules leads to the best performance.

Furthermore, to validate the facilitating effect of the external feature construction method mentioned in Section 3.2, we compare the performance of five different methods: *IMGAUG*, *IMGK*, *IMGCLS*, *EK* and *EFC* in Table 3. To be specific, *IMGAUG* performs contrastive learning using only augmented image features extracted by the CLIP model, without incorporating any external knowledge, which means *IMGAUG* only maximizes the agreement between two augmented views of the same image. In contrast, *EK* and *EFC* incorporate external knowledge into contrastive learning, treating external textual features as additional guidance for the corresponding images. Notably, *EFC* follows the external feature construction method (in Section 3.2) to screen

representative semantic nouns from external knowledge, while *EK* does not apply any such filtering and instead generates external features based on all available nouns. For *IMGK* and *IMGCLS*, non-augmented image features are used as internal visual knowledge. In the case of *IMGK*, the method replaces the external knowledge used in *EK* with internal visual knowledge. Similarly, for *IMGCLS*, this approach replaces the external knowledge used in *EFC* with internal visual knowledge, and ensures that the ratio of clusters to total features is consistent between internal and external knowledge. As illustrated in Table 3, the performance of *EK* is significantly higher compared to *IMGAUG*, *IMGK* and *IMGCLS*. This improvement is particularly evident when evaluated on the MSCOCO dataset, where *EK* demonstrates an average gain of $11.37\%$. Additionally, *EFC* achieves optimal performance across all four datasets. When compared to the *EK* method, the improvement of the *EFC* combination is particularly notable, especially on the Flickr25k dataset. This not only verifies the effectiveness of introducing external knowledge and filtering representative semantic nouns, but also demonstrates the advantages of using external knowledge compared to exhaustively mining internal visual information.

### 4.4. Parameter Analysis

To investigate the impact of the distance threshold $T_1$ and the similarity threshold $T_2$, on performance, we evaluate the MAP results of our proposed DUH-EG method under various values of $T_1$ and $T_2$, with hash codes length equal to 64. As shown in Figure 5, we denote the MAP performance on different datasets using various colors and markers. To be specific, we apply red circles to represent CIFAR-10, blue

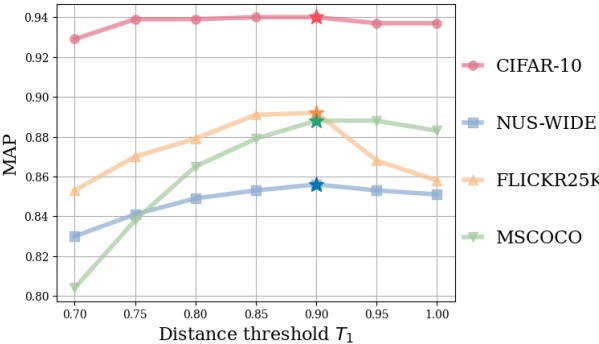

(a) Distance threshold $T_1$ used in Equation (3)

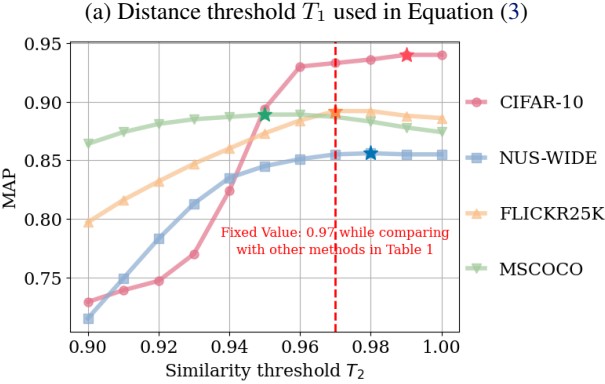

(b) Similarity threshold $T_2$ used in Equation (8)

*Figure 5.* Parameter analysis with 64-bit hash codes on four benchmark datasets.

squares for NUS-WIDE, orange triangles for Flickr25k, and green inverted triangles for MSCOCO. Additionally, we use star markers to indicate the optimal MAP results corresponding to different values of parameters.

To effectively filter the representative semantic nouns based on the extracted text features, we apply a distance threshold $T_1$ to determine the point relatively farther from its cluster center, and treat them as representative features as defined in Equation (3). To further determine the best representative noun screening threshold, we investigate the impact of the distance threshold $T_1$ in Figure 5a. The results show that the optimal similarity threshold is 0.9 on four benchmark datasets. To enhance the effectiveness of the InfoNCE loss, we apply a similarity threshold $T_2$ to distinguish positive and negative sample pairs utilized in the InfoNCE loss (defined in Equation (8)). As illustrated in Figure 5b, we also investigate the impact of the similarity threshold $T_2$. Since the cosine similarity between each pair of external features is concentrated at higher values, even slight variations in $T_2$ can significantly influence the determination of false negative pairs in contrastive learning. Therefore, we measured the optimal threshold value within a narrow range from 0.9 to 1.0 to ensure the precise selection of these pairs. It can be observed that the optimal values of $T_2$ for the CIFAR-10, NUS-WIDE, Flickr25k, and MSCOCO datasets are 0.99,

0.98, 0.97, and 0.95, respectively. For consistency while comparing with other approaches (as shown in Table 1), we set $T_2$ to a common value of 0.97 across all datasets. This setting is indicated by the red vertical line in Figure 5b.

## 5. Conclusion

In this paper, we present a novel deep unsupervised hashing method that integrates external textual information as semantic guidance to address the limitations of inherent visual structures and enhance discrete representation learning. Unlike existing works, our method offers two key contributions: i) To prevent excessive synonyms from dominating the semantic features, we propose a simple yet effective method for selecting representative semantic nouns, thereby ensuring diversity and discriminability of external textual features. ii) We present a bidirectional contrastive learning loss that maximizes the alignment between semantic positive pairs, rather than just between views of the same image, improving robustness and effectiveness. By incorporating external textual information as guidance, we improve matching similarity among images, facilitating more accurate hash learning. We validate the effectiveness of our method through extensive experiments on four benchmark datasets, demonstrating its superior performance compared to 14 state-of-the-art approaches.

## Acknowledgments

This work was supported in part by the National Natural Science Foundation of China (NSFC) under Grants 62472295, 62176171, and U24B20174; the Fundamental Research Funds for the Central Universities under Grants CJ202303 and CJ202403; the Sichuan Science and Technology Planning Project under Grant 24NSFTD0130; the TCL Science and Technology Innovation Fund; the EDB Space Technology Development Programme under Project S22-19016-STDP; and the SCU-LuZhou Science and Technology Cooperation Program under Grant 2023CDLZ-16.

## Impact Statement

This paper proposes a method that aims to advance hashing image retrieval by integrating external text knowledge from WordNet. This approach is evaluated on publicly available image datasets that do not pose security or privacy risks. However, the performance of our method is partially dependent on the distribution of both the training data and the external knowledge. If there exists a domain shift between them, the model's effectiveness may be compromised. Therefore, to enhance its reliability in specific application domains like medicine and biology, it is advisable to introduce external textual knowledge that closely aligns with the target domain.

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

# A. Additional Experiment

## A.1. Visualization

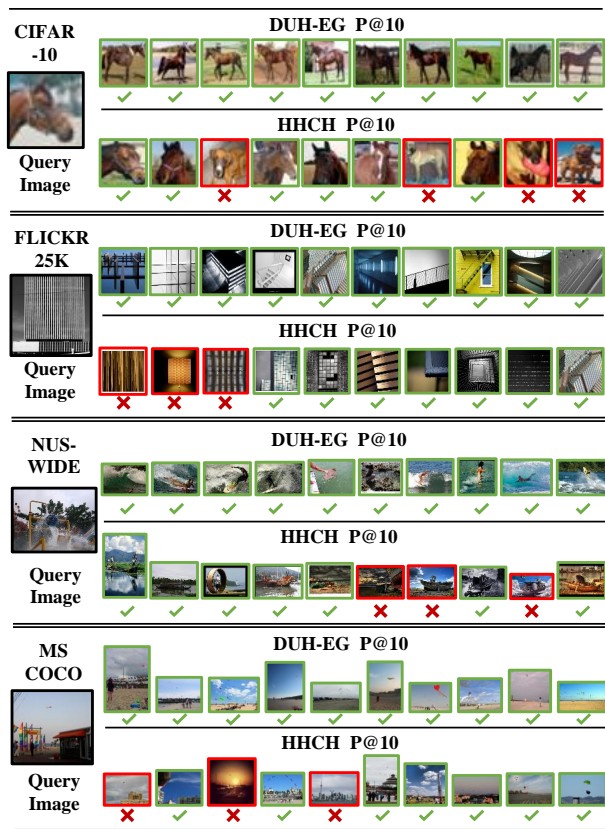

*Figure 6.* The Top10 retrieval results of our proposed DUH-EG and HHCH (Wei et al., 2024) on four benchmark datasets with 64 bits hash codes. The red box indicates the incorrect retrieval results, while the green box represents correct ones. Note that for multi-label datasets Flickr25k, NUS-WIDE and MSCOCO, a correct retrieval result indicates that the retrieved image shares at least one label with the query image.

We illustrate the Top10 retrieval results on CIFAR-10, NUS-WIDE, Flickr25k and MSCOCO datasets with 64 bits hash codes in Figure 6, respectively. On four benchmark datasets, our DUH-EG outperforms the HHCH hashing method (Wei et al., 2024), showing a higher number of correct samples in the Top10 retrieval list. Moreover, we visualize the Top5 nouns matched by two different screening methods corresponding to the same image in Figure 7. As defined in Section 4.3, we filter representative semantic nouns through the *EFC* method and match them with image data, as described in Section 3.2. From Figure 7, it can be observed that, in the absence of the *EFC* method, the Top5 matched nouns include "Round shape", "Ovoid", "Hair ball", "Cat sleep" and "Back circle". These terms primarily focus on shape-related attributes of the image content. Such an em-

phasis not only induces redundancy in the external features but also attenuates the weight associated with the primary discriminative term ("cat sleep"). This leads to an excessive allocation of weight to the shape feature when constructing the external features of the image by Equation (5), rather than to the object itself. Conversely, when the *EFC* method is employed, the Top5 matched nouns highlight specific objects relevant to the image and accurately capture the essence of the depicted scene. Therefore, by utilizing the *EFC* method, more precise external features can be constructed.

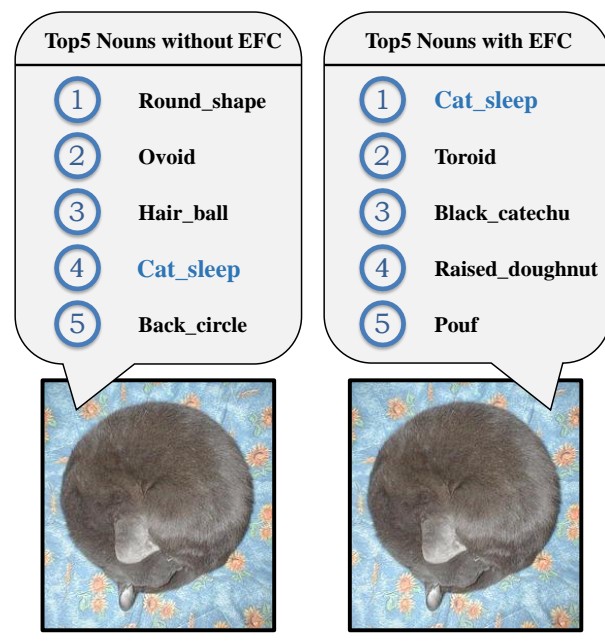

*Figure 7.* The Top5 matched external nouns (with the Top5 softmax probability computed by Equation (6)) corresponding to an image from the Flickr25k dataset. The left panel presents the Top5 nouns without using the *EFC* method defined in Section 4.3, while the right panel displays the Top5 nouns obtained when the *EFC* method is applied. We highlight the noun that best matches the given image in blue.

## A.2. Performance

| Vocabulary | Word Num | Pre-processing Time |
|---|---|---|
| ImageNet | 21,843 | 99.53s |
| WordNet | 117,797 | 476.79s |
| ConceptNet | 134,364 | 531.12s |
| GloVe | 317,756 | 1182.11s |

*Table 4.* The number of words and pre-processing time for different external knowledge sources.

| Vocabulary | CIFAR-10 | | | NUS-WIDE | | | Flickr25k | | | MSCOCO | | |
|---|---|---|---|---|---|---|---|---|---|---|---|---|
| Set | 16 bits | 32 bits | 64 bits | 16 bits | 32 bits | 64 bits | 16 bits | 32 bits | 64 bits | 16 bits | 32 bits | 64 bits |
| ImageNet | 0.940 | 0.942 | 0.941 | 0.849 | 0.854 | 0.855 | 0.871 | 0.885 | 0.888 | 0.864 | 0.885 | 0.892 |
| WordNet | 0.939 | 0.940 | 0.940 | 0.849 | 0.856 | 0.856 | 0.874 | 0.887 | 0.892 | 0.862 | 0.881 | 0.888 |
| ConceptNet | 0.938 | 0.940 | 0.940 | 0.850 | 0.853 | 0.855 | 0.875 | 0.885 | 0.890 | 0.865 | 0.886 | 0.893 |
| GloVe | 0.932 | 0.936 | 0.935 | 0.846 | 0.852 | 0.852 | 0.870 | 0.882 | 0.887 | 0.842 | 0.863 | 0.869 |

*Table 5.* The MAP performance while using different external vocabulary sets.

| Pre-trained | CIFAR-10 | | | NUS-WIDE | | | Flickr25k | | | MSCOCO | | |
|---|---|---|---|---|---|---|---|---|---|---|---|---|
| Model | 16 bits | 32 bits | 64 bits | 16 bits | 32 bits | 64 bits | 16 bits | 32 bits | 64 bits | 16 bits | 32 bits | 64 bits |
| CLIP (ViT-B/16) | 0.939 | 0.940 | 0.940 | 0.849 | 0.856 | 0.856 | 0.874 | 0.887 | 0.892 | 0.862 | 0.881 | 0.888 |
| CLIP (ViT-B/32) | 0.920 | 0.929 | 0.920 | 0.855 | 0.856 | 0.860 | 0.869 | 0.887 | 0.888 | 0.848 | 0.868 | 0.887 |

*Table 6.* The MAP performance while using different pre-trained models.

To evaluate the computational cost and scalability of our approach, we conducted experiments using external vocabulary sets of varying sizes (as shown in Table 4), derived from four representative sources: (1) noun category labels from ImageNet; (2) nouns extracted from WordNet; (3) noun entries from the ConceptNet knowledge graph; and (4) the full vocabulary from GloVe embeddings. Table 4 reports the pre-processing time required by our method for each of the above sources. As our approach relies on CLIP to extract textual features for all words in the vocabulary, the pre-processing time increases substantially with vocabulary size. This observation underscores the importance of exploring more efficient textual feature extraction strategies, which we identify as a promising direction for future work. It is important to note that once the external knowledge is pre-processed, both training and inference stages proceed identically to baseline methods, incurring no additional computational overhead.

Moreover, to assess scalability, we further evaluated model performance across the same four external vocabulary sources, with results presented in Table 5. Our External Feature Construction (EFC) module (see Section 3.2) effectively reduces redundancy in external textual features, maintaining consistent performance across diverse external sources. These results demonstrate the robustness and scalability of our method in handling varying degrees of external knowledge integration.

Additionally, our method leverages Bidirectional Contrastive Learning (in Section 3.3) to align textual and visual representations of images, making the quality of external knowledge introduced via the textual modality a key factor in the overall performance of the hashing network. To investigate this, we trained our model using textual features from two different pre-trained CLIP models: ViT-B/16 and ViT-B/32. As shown in Table 6, the MAP performance on

CIFAR-10 and MSCOCO datasets varies significantly between the two variants, confirming the significant impact of the pre-trained model choice. These findings suggest that similar effects can be expected when adopting other multimodal backbones, as the selection of the textual encoder directly influences the semantic richness of the extracted external text features and thus the retrieval effectiveness of the proposed framework.

## B. Optimization Procedure

To present the method flow introduced in Section 3 more clearly and systematically, we provide a detailed description of the DUH-EG learning process in Algorithm 1 and Algorithm 2. In Algorithm 1 we detail the construction of external features for each image, while Algorithm 2 illustrates the Bidirectional Contrastive Learning process.

---

**Algorithm 1** The process of Section 3.2

---

**Input:** Training images $\mathcal{X} = \{x_i\}_{i=1}^{N_i}$, External nouns $\mathcal{N} = \{n_w\}_{w=1}^{N_t}$, Text encoder $E_t$, Image encoder $E_i$, Distance threshold $T_1$.
```
/* - E_t, E_i:  Encoders from the
    pre-trained CLIP model.          */
```
**Output:** Constructed external features $\mathcal{E} = \{e_i\}_{i=1}^{N_i}$.

1 Construct natural phrases: $n_w \rightarrow \{text_w^p\}_{p=1}^{N_p}$;
```
/* where N_p = 7 in practice.         */
```
2 Extract textual features: $t_w^p = E_t(text_w^p)$;
3 Generate noun features $\bar{t}_w$ by Equation (2);
4 Perform clustering: K-means$(\bar{t}_w) \rightarrow \{c_k\}_{k=1}^{N_k}$;
```
/* c_k means the cluster center of the
    k-th cluster.                     */
```
5 **for** $w = 1$ *to* $N_t$ **do**
```
    /* Denote t̄_w as t̄_{w,k} if t̄_w ∈ k-th
        cluster.                      */
```
6      Compute distance: $dis(\bar{t}_{w,k}, c_k)$;
```
    /* - dis(·,·):  A function measuring
        normalized distance between
        vectors.                      */
```
7      **if** $dis(\bar{t}_{w,k}, c_k) > T_1$ **then**
8          $\bar{t}_w \in \{\bar{t}_w^{rep}\}_{w=1}^{N_{rep}}$;
9      **else**
10          $\bar{t}_w \in \{\bar{t}_w^{nor}\}_{w=1}^{N_{nor}}$;

11 Compute the softmax probability of $\bar{t}_w^{nor}$ corresponding to each cluster center $p_{w,k} = S(\bar{t}_w^{nor}, c_k)$ by Equation (4);
```
/* - S:  The softmax function used to
    compute probability distributions.
    */
```
12 Assign $\{\bar{t}_w^{cen}\}_{w=1}^{N_{cen}}$: $\bar{p}_{w,k}(y = k) = sort\{p_{w,k}(y = k|\bar{t}_w^{nor})| \arg\max p_{w,k}(y|\bar{t}_w^{nor}) = k\}[1]$;
```
/* If S(t̄_w^{nor}, c_k) → p̄_{w,k}(y = k), t̄_w^{nor} ∈ {t̄_w^{cen}}_{w=1}^{N_{cen}}
    */
```
13 Assign representative semantic noun subset: $\{\bar{t}_w^{cen}\}_{w=1}^{N_{cen}} \cup \{\bar{t}_w^{rep}\}_{w=1}^{N_{rep}} \rightarrow \{\bar{t}_w^{com}\}_{w=1}^{N_{com}}$;
14 Extract image features: $i_i = E_i(x_i)$;
15 Construct external features $e_i = \sum_{w=1}^{N_{com}}$ by Equation (5).

---

**Algorithm 2** The process of Section 3.3

---

**Input:** Training images $\mathcal{X} = \{x_i\}_{i=1}^{N_i}$, External features constructed in Algorithm 1 $\mathcal{E} = \{e_i\}_{i=1}^{N_i}$, Image hash function $\mathcal{H}_v$, Text hash function $\mathcal{H}_e$, Similarity threshold $T_2$, Training epochs $\mathcal{T}$.
**Output:** Optimized hash function $\mathcal{H}_v$.

1 *Initialize* $\Theta_v$ *and* $\Theta_e$ *randomly;*
2 **for** $i = 1$ *to* $N_i$ **do**
3      **for** $j = 1$ *to* $N_i$ **do**
4          Compute similarity: $cos(e_i, e_j)$;
```
        /* - cos(·,·):  A similarity
            function measuring the cosine
            similarity between vectors.  */
```
5          **if** $cos(e_i, e_j) \leq T_2$ **then**
6             treat the $j$-th sample as forming a negative pair with the $i$-th sample;
7          **else**
8             treat the $j$-th sample as forming a positive pair with the $i$-th sample;

9 **for** $t = 1$ *to* $\mathcal{T}$ **do**
10      Generate two differently augmented views of images: $\mathcal{V}_1 = \{v_i^1\}_{i=1}^{N_i}$ and $\mathcal{V}_2 = \{v_i^2\}_{i=1}^{N_i}$;
11      Generate hash codes: $h_i^{v_1} = \mathcal{H}_v(v_i^1, \Theta_v)$, $h_i^{v_2} = \mathcal{H}_v(v_i^2, \Theta_v)$ and $h_i^e = \mathcal{H}_e(e_i, \Theta_e)$ ;
12      Compute $\mathcal{L}_{BCL}$ with Equation (1);
13      Update $\Theta_v$ and $\Theta_e$ with the Adam optimizer.

---

