# OpenReview forum: "Deep Unsupervised Hashing via External Guidance"
_ICML.cc/2025/Conference — ICML 2025 poster_

### Official Review · Reviewer_m7gz · 2025-03-09

**Overall Recommendation:** 4

**Summary:**

This paper introduces DUH-EG, a deep unsupervised hashing framework that integrates external textual information as semantic guidance to overcome the limitations of relying solely on internal visual structures. The method first constructs external features by extracting and clustering textual features (derived from WordNet nouns and a pre-trained CLIP model) to select representative semantic nouns. It then employs a bidirectional contrastive learning loss to align hash codes generated from two augmented views of an image and its corresponding external textual view. Extensive experiments on standard image retrieval benchmarks (CIFAR-10, NUS-WIDE, FLICKR25K, and MSCOCO) demonstrate that DUH-EG outperforms several state-of-the-art unsupervised hashing methods.

**Claims And Evidence:**

Yes, the claims made in the submission are supported by clear and convincing evidence. The paper claims that incorporating external textual guidance can enrich semantic supervision and lead to more discriminative hash codes. These claims are supported by the comprehensive experimental results, such as comparison experiments, ablation studies, and parameter analysis.

**Essential References Not Discussed:**

No. This paper have cited the essential related works.

**Experimental Designs Or Analyses:**

Yes, I checked the soundness of the experimental designs and analyses. The experimental design is robust:
1) The authors evaluate on multiple benchmark datasets.
2) Comparisons are made against 14 state-of-the-art unsupervised hashing methods.
3) Ablation studies and parameter sensitivity analyses provide insights into the contribution of each component.

**Methods And Evaluation Criteria:**

Yes, both the proposed methods and evaluation criteria make sense for the problem at hand. Specifically, the proposed method is well motivated and clearly described. Leveraging a pre-trained CLIP model to extract and fuse textual features, followed by a clustering-based selection mechanism, is a creative strategy to overcome synonym redundancy. The use of a bidirectional contrastive loss—extending traditional contrastive learning to align not only different augmented views but also external textual views—is appropriate for enhancing semantic consistency. Evaluation metrics such as MAP, Precision-Recall curves, and TopN-Precision are standard and suitable for assessing image retrieval performance.

**Other Comments Or Suggestions:**

1) A more detailed analysis of computational overhead would be valuable, especially when scaling to larger external noun databases.
2) Consider discussing potential impacts of noisy or less relevant textual features.
3) Minor editorial improvements and clarifications in the description of the bidirectional contrastive loss could help improve readability.

**Other Strengths And Weaknesses:**

Strengths:
1) Novel integration of external textual guidance with unsupervised hashing.
2) Comprehensive experimental validation across multiple benchmark datasets.
3) Clear ablation studies and parameter analyses that illustrate the effectiveness of individual components.

Weaknesses:
1) Limited discussion on the scalability and computational cost of extracting and processing external textual features.
2) Lack of deeper analysis on why external guidance particularly improves the discrimination of hash codes.
3) Some methodological details (e.g., sensitivity to the choice of pre-trained models) could be elaborated further.

**Questions For Authors:**

1) How does DUH-EG scale when the external textual database is significantly larger, and what are the computational implications?
2) Have you evaluated the impact of noisy or irrelevant external textual features on the hash code quality?
3) Can you provide further justification for the selection of the similarity thresholds (T1 and T2) across different datasets? Are these hyperparameters sensitive to dataset characteristics?
4) How does the choice of pre-trained models (e.g., using alternatives to CLIP) affect performance? Would similar improvements be expected with other multi-modal models?

**Relation To Broader Scientific Literature:**

The paper is well situated within the broader context of unsupervised hashing and contrastive learning. It builds on recent advances in using internal data structures for hash code learning while making a novel contribution by integrating external textual information. The work is compared with and references several key studies in both unsupervised hashing and multi-modal representation learning, clearly delineating its contributions relative to prior art.

**Theoretical Claims:**

There is no any proofs for theoretical claims. The paper primarily focuses on algorithmic and empirical contributions rather than deep theoretical proofs. The derivation of the bidirectional contrastive loss and the rationale for the external feature construction are presented clearly. The formulation is standard within the contrastive learning literature and appears correct.

---

> ### Author Rebuttal · Authors · 2025-04-01
>
> We sincerely appreciate your insightful comments and suggestions. To enhance readability, we will refine the description of the bidirectional contrastive loss in the revised manuscript accordingly.
>
> **Q1: How does DUH-EG scale when the external textual database is significantly larger, and what are the computational implications?**
>
> **R1:** Our method enhances scalability by selecting representative textual features and aggregating them into a single external feature **e** using Eq.(5) of the original paper. This ensures scalability when handling a significantly larger textual database.
>
> Regarding computational cost and scalability, Table 1 and Table 2 analyze computational cost, showing that while pre-processing time increases with larger external noun databases, MAP performance remains stable, demonstrating scalability. Additionally, the MAP performance remains largely unchanged, indicating that our method exhibits good scalability. However, in the inference phase, since only the image hashing network is used to generate discriminative hash codes, the size of the textual database has no impact on inference efficiency.
>
> ---
>
> **Q2: Have you evaluated the impact of noisy or irrelevant external textual features on the hash code quality?**
>
> **R2:** Yes. We highlighted the positive impact of filtering redundant external textual features using the External Feature Construction (EFC) method (Section 3.2) in the supplementary material. As shown in Fig. 7, after filtering, the feature "Cat_sleep" had the highest weight. Without EFC, irrelevant features like "Round_shape," "Ovoid," and "Hair_ball" had higher weights than "Cat_sleep." The ablation study in Table 3 confirms that irrelevant features degrade the quality of the learned hash codes.
>
> **Q3: Can you provide further justification for the selection of the similarity thresholds $T_1$ and $T_2$ across different datasets? Are these hyperparameters sensitive to dataset characteristics?**
>
> **R3:**
> 1. To ensure consistency across datasets, we fixed $T_1$ at 0.90 and $T_2$ at 0.97 when comparing with other methods in Table 1 of the original paper.
> 2. No, they are not sensitive to datasets, so we use the same parameters in all datasets.
>
> **Q4: How does the choice of pre-trained models (e.g., using alternatives to CLIP) affect performance? Would similar improvements be expected with other multi-modal models?**
>
> **R4:**
> 1. Since our method uses Bidirectional Contrastive Learning to align an image's textual and visual representations, the quality of external knowledge introduced through the textual modality affects the hashing network’s performance. To validate this, we trained our model with textual features from CLIP (ViT-B/16) and CLIP (ViT-B/32). As shown in Table 3, the MAP performance differs significantly on CIFAR-10 and MSCOCO, confirming the impact of the pre-trained model choice.
> 2. Yes, similar improvements would be expected with other multi-modal models, as evidenced by the results of CLIP (ViT-B/16) and CLIP (ViT-B/32).
>
> **Table 1: The number of words and pre-processing time for different external knowledge.**
>
> | External Knowledge | Word Num | Pre-processing Time |
> |--------------------|----------|---------------------|
> | ImageNet           | 21,843   | 99.53s              |
> | WordNet            | 117,797  | 476.79s             |
> | ConceptNet         | 134,364  | 531.12s             |
> | GloVe              | 317,756  | 1182.11s            |
>
> **Table 2: The MAP performance while using different external knowledge.**
>
> | External Knowledge | CIFAR-10 (16, 32, 64 bits) | NUS-WIDE (16, 32, 64 bits) | FLICKR25K (16, 32, 64 bits) | MSCOCO (16, 32, 64 bits) |
> |--------------------|---------------------------|----------------------------|----------------------------|--------------------------|
> | ImageNet           | 0.940 / 0.942 / 0.941     | 0.849 / 0.854 / 0.855      | 0.871 / 0.885 / 0.888      | 0.864 / 0.885 / 0.892    |
> | WordNet            | 0.939 / 0.940 / 0.940     | 0.849 / 0.856 / 0.856      | 0.874 / 0.887 / 0.892      | 0.862 / 0.881 / 0.888    |
> | ConceptNet         | 0.938 / 0.940 / 0.940     | 0.850 / 0.853 / 0.855      | 0.875 / 0.885 / 0.890      | 0.865 / 0.886 / 0.893    |
> | GloVe              | 0.932 / 0.936 / 0.935     | 0.846 / 0.852 / 0.852      | 0.870 / 0.882 / 0.887      | 0.842 / 0.863 / 0.869    |
>
> **Table 3: The MAP performance while using different pre-trained models.**
>
> | Pre-trained Model   | CIFAR-10 (16, 32, 64 bits) | NUS-WIDE (16, 32, 64 bits) | FLICKR25K (16, 32, 64 bits) | MSCOCO (16, 32, 64 bits) |
> |---------------------|---------------------------|----------------------------|----------------------------|--------------------------|
> | CLIP (ViT-B/16)     | 0.939 / 0.940 / 0.940     | 0.849 / 0.856 / 0.856      | 0.874 / 0.887 / 0.892      | 0.862 / 0.881 / 0.888    |
> | CLIP (ViT-B/32)     | 0.920 / 0.929 / 0.920     | 0.855 / 0.856 / 0.860      | 0.869 / 0.887 / 0.888      | 0.848 / 0.868 / 0.887    |

---

### Official Review · Reviewer_w54n · 2025-03-09

**Overall Recommendation:** 4

**Summary:**

This paper proposes a deep unsupervised hashing framework (DUH-EG) designed for image retrieval. Unlike traditional unsupervised hashing methods that rely solely on intrinsic visual structures, DUH-EG leverages external textual guidance extracted from a lexical database (i.e., WordNet) and processed via a pre-trained CLIP model. The method comprises two components: (1) an external feature construction module that selects representative semantic nouns using clustering and filtering, and (2)) a bidirectional contrastive learning loss that aligns hash codes from two augmented views of an image with its corresponding external textual feature. Extensive experiments on four widely-used benchmarks (i.e., CIFAR-10, NUS-WIDE, FLICKR25K, and MSCOCO) show considerable improvements in MAP over state-of-the-art approaches.

**Claims And Evidence:**

Yes

**Essential References Not Discussed:**

None

**Experimental Designs Or Analyses:**

1. The authors evaluate DUH-EG on four standard benchmarks, comparing against 14 state-of-the-art unsupervised hashing methods. The comprehensive comparison and the use of diverse backbones (i.e., VGG-16 and CLIP) strengthen the empirical study.
2. Detailed ablation experiments dissect the contributions of individual modules (i.e., SIM and AUG) and various configurations (e.g., using internal visual knowledge vs. external textual knowledge). The parameter analysis on thresholds T1 and T2 further validates the robustness of the proposed method.
3. The experimental analysis is thorough. However, a discussion regarding the computational overhead introduced by the external feature extraction and the sensitivity to pre-trained models (like CLIP) would provide additional clarity.

**Methods And Evaluation Criteria:**

1. This paper introduces a two-phase approach. In the first phase, textual features for nouns are fused and clustered to select a diverse set of representative semantic features. In the second phase, these external features are aligned with internal image representations via a specially designed bidirectional contrastive loss. This loss not only maximizes mutual information between different augmented views but also redefines positive pairs by using external guidance to mitigate false negatives.
2. The evaluation is primarily based on MAP metrics across four image retrieval datasets. In addition, the authors provide ablation studies, sensitivity analyses for the key thresholds (T1 and T2), and supplementary metrics (P-R and P@N curves) to reinforce the empirical performance.

**Other Comments Or Suggestions:**

Typo: Page 6, "token the remaining images" -> "took the remaining images".

**Other Strengths And Weaknesses:**

Strengths:
1. The idea of using external textual guidance to overcome intrinsic limitations of visual data is novel and interesting.
2. Extensive experiments with competitive baselines across multiple datasets demonstrate the efficacy of the proposed method.
3. The detailed ablation studies and parameter analyses provide a clear breakdown of the contributions of each component.
4. The method and experiments are described with sufficient clarity, making the approach understandable.

Weaknesses:
1. A discussion on the computational costs and scalability issues associated with integrating external guidance would be beneficial.
2. Relying on external pre-trained models (e.g., CLIP) and WordNet might limit the method's applicability in scenarios where such resources are constrained.

**Questions For Authors:**

1. Why was T2 fixed at 0.97 for all datasets despite varying optimal values (Fig. 3b)? What about performance if dataset-specific T2 is used?
2. What is the training/inference time of DUH-EG compared to baselines, especially with CLIP?
3. Will code and pretrained models be released?

**Relation To Broader Scientific Literature:**

The paper positions its contributions well within the literature on unsupervised hashing, contrastive learning, and multimodal representation learning. It advances the field by combining cross-modal external guidance with contrastive learning, addressing a known limitation of internal visual structures.

**Theoretical Claims:**

There is no theoretical claim.

---

> ### Author Rebuttal · Authors · 2025-04-01
>
> We sincerely appreciate your valuable feedback and constructive suggestions. Below are our point-to-point responses:
>
> **Q1: A discussion on the computational costs and scalability issues.**
>
> **R1:** To analyze computational costs and scalability, we evaluated our approach using external vocabulary sets of varying sizes from four sources: 1) Noun category labels from ImageNet; 2) Nouns extracted from WordNet; 3) Nouns from the ConceptNet knowledge graph; and 4) The full vocabulary from GloVe.
>
> The experimental results are shown in Table 1, presenting the pre-processing time of our method for each external knowledge source. Since our approach extracts textual features for all words using CLIP, the pre-processing time increases as the vocabulary size grows significantly. Therefore, exploring more efficient strategies for extracting external textual features is a promising direction for our future research. Notably, after external knowledge preprocessing, our training and inference are the same as the existing methods with no additional time added.
>
> Regarding scalability, we evaluated the model performance using four external vocabulary sources with different sizes, as shown in Table 2. Our External Feature Construction (EFC) method (in section 3.2 of the original paper) effectively mitigates redundancy in external textual features, ensuring robust performance even when different external knowledge sources are used. This highlights our method's adaptability and scalability.
>
> **Q2: Relying on external pre-trained models (e.g., CLIP) and WordNet might limit the method's applicability in scenarios where such resources are constrained.**
>
> **R2:** Our method leverages the pre-trained CLIP to incorporate textual features from WordNet as external knowledge for unsupervised hashing. However, we would like to clarify that our approach does not strictly rely on these specific resources. In resource-constrained scenarios, lightweight or domain-specific pre-trained multimodal models and textual knowledge bases could be used as alternatives.
>
> **Q3: Why was $T_2$ fixed at 0.97 for all datasets despite varying optimal values (Fig. 3b)?**
>
> **R3:** To maintain a consistent set of hyperparameters across all datasets, we fixed $T_2=0.97$ in comparison experiments, thereby reducing parameter tuning costs. If dataset-specific $T_2$ values were used, the performance would improve slightly, as indicated by the star-marked points in Fig. 3b.
>
> **Q4: What is the training/inference time of DUH-EG compared to baselines, especially with CLIP?**
>
> **R4:** We freeze the parameters of CLIP and train only the subsequent hashing layers. This makes our training process more efficient than updating all parameters of CLIP. For inference, our method consists of two steps: (1) feature extraction using CLIP and (2) hashing the features via the trained hashing layers. Moreover, the hashing layers consist of only two fully connected layers [$F$--$512$--$L$] with activation functions, which introduce negligible additional overhead (about 0.3M params). We provide a detailed overview of the computational costs associated with our method in Table 3, which demonstrates the efficiency of our method compared with CLIP.
>
> **Q5: Will code and pretrained models be released?**
>
> **R5:** Yes. We will release them after the acceptance of the paper.
>
>
> **Table 1: The number of words and pre-processing time for different external knowledge.**
> | External Knowledge | Word Num | Pre-processing Time |
> |------------------------|-------------|-------------------------|
> | ImageNet | 21,843 | 99.53s |
> | WordNet | 117,797 | 476.79s |
> | ConceptNet | 134,364 | 531.12s |
> | GloVe | 317,756 | 1182.11s |
>
>
> **Table 2: The MAP performance using different external knowledge.**
> | External Knowledge | CIFAR-10 (16, 32, 64 bits) | NUS-WIDE (16, 32, 64 bits)| FLICKR25K (16, 32, 64 bits) | MSCOCO (16, 32, 64 bits) |
> |------------------------|--------------------------------|--------------------------------|--------------------------------|------------------------------|
> | ImageNet | 0.940 / 0.942 / 0.941 | 0.849 / 0.854 / 0.855 | 0.871 / 0.885 / 0.888 | 0.864 / 0.885 / 0.892 |
> | WordNet | 0.939 / 0.940 / 0.940 | 0.849 / 0.856 / 0.856 | 0.874 / 0.887 / 0.892 | 0.862 / 0.881 / 0.888 |
> | ConceptNet | 0.938 / 0.940 / 0.940 | 0.850 / 0.853 / 0.855 | 0.875 / 0.885 / 0.890 | 0.865 / 0.886 / 0.893 |
> | GloVe | 0.932 / 0.936 / 0.935 | 0.846 / 0.852 / 0.852 | 0.870 / 0.882 / 0.887 | 0.842 / 0.863 / 0.869 |
>
> **Table 3: Training/inference time, FLOPs, and params of DUH-EG and CLIP on an NVIDIA 3090 GPU.**
> | Model | Training Time / epoch | Inference Time / sample| GFLOPs| Params|
> |------------|----------------------------------------------------|----------------------------|------------------|-------------|
> | DUH-EG | 105.27s (without pre-processing of external knowledge) | 4.1227s | 11.2703 | 57.56M |
> | CLIP | The CLIP was not trained on the 3090 GPU | 4.1223s | 11.2700 | 57.26M |

---

### Official Review · Reviewer_vrcG · 2025-03-11

**Overall Recommendation:** 4

**Summary:**

The paper identifies a crucial bottleneck in unsupervised image hashing, i.e., the limitation of insufficient knowledge guidance solely relying on the visual structures. To remedy this, semantic representatives are selected from external textual databases, serving as external guidance for the image modality via a bidirectional contrastive loss. Leveraging textual external knowledge, they designed a novel contrastive loss to avoid false negative pairs. Experiments on various datasets and model architectures have validated the consistent effectiveness of the method in mining and preserving semantic similarity for unsupervised hashing.

## update after rebuttal
The authors' rebuttal adequately addressed these concerns. As a result, I keep my score as 4 (recommend acceptance).

**Claims And Evidence:**

Yes. Claims regarding proposed approaches are well-verified by extensive ablation results w.r.t. all 4 evaluated datasets and multiple hash code lengths.

**Essential References Not Discussed:**

Key references are well-discussed.

**Experimental Designs Or Analyses:**

The paper’s method is verified comprehensively and analyzed thoroughly via extensive experiments to establish its validity.

**Methods And Evaluation Criteria:**

The proposed method is clear and reasonable with unique consideration of external textual characteristics. The evaluation criteria are also reasonable as the evaluated datasets vary in scales and the adopted evaluation metrics are widely-used for the hashing task.

**Other Comments Or Suggestions:**

1. The paper is clear in its language but it’s better to check some mathematical operations with different names (e.g., 'sim' and 'cos' in Eq.(3),(4),(6),(8), and Line 256-258) which may have the same meanings and cause ambiguity.
2. The authors may limit the use of boldface abbreviations only for necessary cases.

**Other Strengths And Weaknesses:**

Strengths:
1. As summarized, the authors identify a knowledge bottleneck in unsupervised image hashing retrieval. The proposed method is well-motivated and novel by analyzing of the key challenges and potential problems from learning with external guidance.
2. The experiments compare state-of-the-art methods with datasets of different scales and complexities, while presenting extensive analyses on both the learning approaches and the external feature construction.
3. The paper effectively presents the proposed method with generally excellent clarity.

Weaknesses:
1. More recent works could be included in the related work of deep unsupervised hashing.

**Questions For Authors:**

1. Why does the similarity threshold T_1 specifically affect the Flickr25k dataset when its values is high?
2. Why does a reduction in T_2 have a more significant impact on the model compared to T_1?

**Relation To Broader Scientific Literature:**

The work is a novel improvement in unsupervised semantic image hashing, while being potential to be applied in hashing retrieval of other modalities that can be associated with textual knowledge. The paper’s method is closely related to contrastive learning and clustering approaches.

**Theoretical Claims:**

The paper does not contain proofs/theoretical claims.

---

> ### Author Rebuttal · Authors · 2025-04-01
>
> We sincerely appreciate your valuable feedback and constructive suggestions. Below are our point-to-point responses:
>
> **Q1: More recent works could be included in the related work of deep unsupervised hashing.**
>
> **R1:** Thank you for your suggestion. We will update the related work section to incorporate more recent studies on deep unsupervised hashing, ensuring a more comprehensive and up-to-date literature review.
>
> ---
>
> **Q2: The paper is clear in its language, but it’s better to check some mathematical operations with different names (e.g., 'sim' and 'cos' in Eq.(3), (4), (6), (8), and Lines 256-258) which may have the same meanings and cause ambiguity.**
>
> **R2:** We appreciate your careful examination of our mathematical operations. To eliminate potential ambiguity, we will revise the notation in Eq.(3), (4), (6), and (8) of the original paper, as well as in Lines 256–258, ensuring consistency in mathematical operations.
>
> ---
>
> **Q3: The authors may limit the use of boldface abbreviations only for necessary cases.**
>
> **R3:** Thanks for your suggestion. We will refine the formatting to use boldface abbreviations only where necessary, improving readability and clarity.
>
> ---
>
> **Q4: Why does the similarity threshold $T_1$ specifically affect the Flickr25k dataset when its value is high?**
>
> **R4:** The similarity threshold $T_1$ plays a crucial role in selecting representative textual features with lower similarity to the cluster center. As shown in Eq.(5) of the original paper, the final fused external feature e is computed from the external textual feature set $\left(\bar{\{t}}_w^{com}\right)$. Thus, reducing redundancy in $\left(\bar{\{t}}_w^{com}\right)$ is essential for ensuring well-matched external textual knowledge for the original images.
>
> If $T_1$ is set too high, some representative textual features may be filtered out during the process described in Eq.(4), weakening the discriminative power of the final fused external feature **e**. Since the representational capacity of **e** is influenced by $\left(\bar{\{t}}_w^{com}\right)$, different datasets may respond differently to variations in $T_1$. In the case of Flickr25k, a higher $T_1$ can significantly impact model performance, likely due to the dataset’s inherent characteristics, making it more sensitive to the removal of certain textual features.
>
> ---
>
> **Q5: Why does a reduction in $T_2$ have a more significant impact on the model compared to $T_1$?**
>
> **R5:** Unlike $ T_1$, which filters representative textual features, the primary role of the similarity threshold $T_2$ is to determine positive and negative sample pairs in Eq.(9). Fluctuations in $T_2$ directly affect the model’s ability to correctly distinguish between potential positive and negative pairs.
>
> Even a slight reduction in $T_2$ can cause some negative sample pairs to be misclassified as positive ones. This misclassification directly affects the discrimination of the learned hash codes, ultimately leading to a decline in overall model performance.

---

### Official Review · Reviewer_2wPY · 2025-03-13

**Overall Recommendation:** 4

**Summary:**

This paper proposes a novel deep unsupervised hashing method, Deep Unsupervised Hashing with External Guidance (DUH-EG), to enhance image retrieval by incorporating external textual knowledge as semantic guidance. The method selects representative semantic nouns from an external textual database, aligns images with them to extract more discriminative external features, and employs a bidirectional contrastive learning mechanism to maximize agreement between hash codes in internal and external spaces. Experiments on CIFAR-10, NUS-WIDE, FLICKR25K, and MSCOCO demonstrate that DUH-EG significantly outperforms state-of-the-art unsupervised hashing methods.

**Claims And Evidence:**

The paper claims that incorporating external textual guidance improves hash learning by overcoming the limitations of internal visual structures. This claim is supported by extensive experiments, where DUH-EG consistently outperforms existing methods across different datasets and hash code lengths.

**Essential References Not Discussed:**

No. The paper has covered a wide range of related works.

**Experimental Designs Or Analyses:**

The experimental design is rigorous, with a comprehensive evaluation across multiple datasets and comparison with state-of-the-art methods. The ablation studies provide insights into the contributions of different proposed components.

**Methods And Evaluation Criteria:**

The proposed methods and evaluation criteria are appropriate for the problem. The paper follows standard benchmarks and metrics, such as Mean Average Precision (MAP), Precision-Recall (P-R) curves, and Top-N Precision. The inclusion of multiple datasets and comparison with strong baselines provides a comprehensive evaluation.

**Other Comments Or Suggestions:**

(1) Clarify the selection criteria for external textual features and how they generalize across datasets.
(2) Provide additional ablation studies on the influence of different external knowledge sources.
(3) Discuss potential limitations, such as dependency on pre-trained vision-language models like CLIP.

**Other Strengths And Weaknesses:**

The strengths of the paper include the novelty of integrating external textual knowledge, strong experimental validation, and a well-designed bidirectional contrastive learning mechanism. The primary weakness is the lack of deeper analysis of how different external knowledge sources affect performance.

**Questions For Authors:**

(1) How does the selection of external textual features affect the model's robustness across datasets?
(2) Would using domain-specific external knowledge further enhance performance?
(3) Can this approach be extended to other modalities beyond images and text?

**Relation To Broader Scientific Literature:**

The paper builds upon prior work in deep unsupervised hashing and contrastive learning, while introducing external knowledge as a novel enhancement.

**Theoretical Claims:**

No theoretical claims or proofs are presented; the work is empirically driven.

---

> ### Author Rebuttal · Authors · 2025-04-01
>
> We greatly appreciate your thoughtful feedback and recognition of our contributions. Below are our point-to-point responses:
>
> **Q1: How does the selection of external textual features affect the model's robustness across datasets?**
>
> **R1:** In our work, we focus on widely used benchmarks, such as CIFAR-10, NUS-WIDE, FLICKR25K, and MSCOCO, which primarily contain images of common objects and everyday scenes. To ensure generalization, we use a general vocabulary (e.g., WordNet) as our external knowledge source. Specifically, we extract textual features via CLIP and align them with visual features, promoting semantic consistency among similar images in the learned discrete representations.
>
> However, due to the presence of synonyms, textual features from the general vocabulary often introduce redundancy. To address this, we propose an External Feature Construction (EFC) method (Section 3.2) to cluster semantically similar nouns and select representative textual features, thereby reducing redundancy caused by excessive synonyms. This process enhances the semantic richness of the aggregated external textual features obtained via weighted summation (Eq. 5) and improves their representational capacity.
>
> As demonstrated in our ablation experiments (Table 3), the EFC method improves model performance. Furthermore, as shown in Fig. 7 (Appendix), applying EFC reduces the number of high-weighted nouns with similar meanings, validating its effectiveness in redundancy reduction.
>
> ---
>
> **Q2: Would using domain-specific external knowledge further enhance performance?**
>
> **R2:** Yes, using domain-specific external knowledge can further enhance performance. To evaluate this, we conducted an ablation study using six external knowledge sources (ImageNet, ConceptNet, WordNet_N, WordNet_V, GloVe, and Medical) on the MSCOCO dataset, which belongs to the natural image domain.
>
> - ImageNet represents 21,843 noun labels from the ImageNet dataset.
> - ConceptNet and WordNet_N contain noun data.
> - WordNet_V consists of verb data.
> - GloVe includes words spanning multiple parts of speech.
> - Medical contains vocabulary from the medical domain.
>
> Experimental results in Table 1 show that models trained with general noun vocabulary (e.g., ImageNet, ConceptNet, WordNet_N) perform better, despite differences in vocabulary size. In contrast, models trained on verb-based (WordNet_V) or mixed-POS vocabularies (GloVe) exhibit lower performance, with medical vocabulary yielding the lowest performance.
>
> In summary, for a target dataset (e.g., MSCOCO) in the natural image domain, using domain-relevant external knowledge (e.g., general noun vocabulary) effectively enhances model performance. Thus, matching external knowledge with the target dataset’s domain improves performance.
>
> **Table 1: The ablation studies on the influence of different external knowledge on specific dataset MSCOCO.**
>
> | **External Knowledge** | **Word Num** | **MSCOCO (16 bits, 32 bits, 64 bits)** |
> |------------------------|-------------|----------------------------------------|
> | **Medical**           | 35,500      | 0.807 / 0.835 / 0.846                 |
> | **GloVe**             | 317,756     | 0.842 / 0.863 / 0.869                 |
> | **WordNet_V**         | 11,531      | 0.825 / 0.850 / 0.863                 |
> | **WordNet_N**         | 117,797     | 0.862 / 0.881 / 0.888                 |
> | **ImageNet**          | 21,843      | 0.864 / 0.885 / 0.892                 |
> | **ConceptNet**        | 134,364     | 0.865 / 0.886 / 0.893                 |
>
> **Q3: Can this approach be extended to other modalities beyond images and text?**
>
> **R3:** Yes, our approach could be extended to other modalities, but certain preconditions apply. Specifically, our method aligns external textual features with visual representations via CLIP, making it well-suited for unsupervised image hashing. While extending to other modalities (e.g., audio using models like Wav2CLIP) is promising, its effectiveness depends on the availability of modality-specific external knowledge and corresponding pre-trained models.

---

### Decision · Program_Chairs · 2025-05-01

**Decision:**

Accept (poster)

**Comment:**

This paper introduces a deep unsupervised hashing method, named DUH-EG, which improves image retrieval by incorporating external textual knowledge. It selects key semantic nouns from an external source to guide feature learning and uses bidirectional contrastive learning to align internal and external representations. This dual approach boosts discriminative power and outperforms existing methods on benchmark datasets, which can achieve SOTA retrieval performance. Four reviewers have carefully checked the motivations, technical details, method analysis, and experimental validations. All of our reviewers agree to accept this paper. After the post-rebuttal discussion, the authors’ point-to-point rebuttal addressed most of these concerns so that no significant issues remain. This paper is acceptable in its current form.